# Unveiling the *Re effect* in Ni-based single crystal superalloys

Xiaoxiang Wu [1*], Surendra Kumar Makineni[1*], Christian H. Liebscher[1], Gerhard Dehm [1], Jaber Rezaei Mianroodi [1,2], Pratheek Shanthraj[1,3], Bob Svendsen[1,2], David Bürger[4], Gunther Eggeler[4], Dierk Raabe[1] & Baptiste Gault [1,5*]

Single crystal Ni-based superalloys have long been an essential material for gas turbines in aero engines and power plants due to their outstanding high temperature creep, fatigue and oxidation resistance. A turning point was the addition of only 3 wt.% Re in the second generation of single crystal Ni-based superalloys which almost doubled the creep lifetime. Despite the significance of this improvement, the mechanisms underlying the so-called "Re effect" have remained controversial. Here, we provide direct evidence of Re enrichment to crystalline defects formed during creep deformation, using combined transmission electron microscopy, atom probe tomography and phase field modelling. We reveal that Re enriches to partial dislocations and imposes a drag effect on dislocation movement, thus reducing the creep strain rate and thereby improving creep properties. These insights can guide design of better superalloys, a quest which is key to reducing $CO_2$ emissions in air-traffic.

[1] Max-Planck-Institut für Eisenforschung GmbH, 40237 Düsseldorf, Germany. [2] RWTH Aachen University, 52062 Aachen, Germany. [3] School of Materials, The University of Manchester, M13 9PL Manchester, UK. [4] Institut für Werkstoffe, Ruhr-Universität Bochum, Universitätsstrasse 150, D-44 780 Bochum, Germany. [5] Department of Materials, Royal School of Mine, Imperial College London, London SW7 2AZ, UK. *email: x.wu@mpie.de; makineni@mpie.de; b.gault@mpie.de

One of the limiting factors to further enhancing the efficiency and reliability of modern gas turbine engines is the capability of the material used for engine components to sustain an extremely harsh operation environment, i.e., involving high temperature gradients (700–1200 °C) and severe stresses (100–800 MPa). Decades of research and alloy development have resulted in Ni-based superalloys which are characterised by an excellent combination of resistance to high temperature creep, i.e., to plastic deformation under load at elevated temperature over long periods of time, as well as resistance to high temperature oxidation[1–3].

The excellent mechanical properties of Ni-based superalloys originate from their microstructure. Ni-based superalloys have a characteristic two-phase microstructure comprising a disordered γ matrix phase with a face-centred cubic (FCC) structure and coherently embedded γ′ precipitates that exhibit an FCC-based L1$_2$-ordered structure. Ni-based superalloys often contain ten or more elements in proportions that have been finely tuned to adjust the precipitate shape by misfit engineering, their volume fraction, as well as the overall mechanical properties. Specifically, W and Re are added to strengthen the γ matrix phase. Al, Ta and Ti are used as stabilisers for γ′ precipitates and the elements Cr and Mo for high temperature corrosion and oxidation resistance.

The addition of only 3 wt% Re results in a dramatic improvement in the lifetime of the alloy. It significantly lowers the minimum creep rate that can be reached and the creep-to-rupture ratio can be doubled across different temperature ranges[4–6]. Varying the amount of Re, up to ~6 wt%, also in combination with up to ~ 3 wt% Ru, has led to the development of new generations of superalloys with improved creep properties[7]. These benefits are unfortunately offset by raising the alloy's density[8] and cost, as Re and Ru are expensive rare elements present in trace quantities in the earth's crust. Thus, efforts are being made to find suitable alternatives to replace part or all of the rare elements in the alloy composition without compromising the alloy's original properties[9].

Creep deformation is associated with defects such as dislocations and planar faults, e.g., anti-phase boundaries (APBs), superlattice intrinsic stacking faults (SISFs), superlattice extrinsic stacking faults (SESF) and complex stacking faults (CSFs) in the γ/γ′ microstructure depending on the deformation conditions[10]. These planar faults are generated when mobile dislocations in the γ matrix cut through the γ′ precipitates during creep, which is referred to as the γ′ shearing process. Several recent studies have shown that their local atomic structure is different compared to the surrounding lattice, which is often accompanied by a change in the local composition. Segregation of solutes to dislocations (resulting in Cottrell atmospheres) and to SFs (Suzuki segregation) have been observed by TEM and atom probe tomography (APT) analysis[3,10–18]. As pointed out by Barba et al.[14] and Smith et al.[3,19], depending on the type of solute, the properties of superalloys can be affected by solute segregation, e.g., segregation-assisted plasticity, or phase- transformation-induced strengthening. More specifically, in a single-crystal prototype Ni-based superalloy, Cr and Co enrichment to the APB lowers its energy, facilitating APB-based cutting of γ′[14]. On the other hand, if SFs are decorated by Ta and Nb, local phase transformation (γ′ to η

phase) is favoured, resulting in reduced precipitate shearing[3]. Similar segregation engineering has been used for tailoring mechanical properties of Fe-5 at% Mn maraging steels[20] and medium Mn steels[21,22].

Despite these efforts, the mechanism(s) underlying the influence of Re on creep deformation remain unclear and a topic of on-going debate, hindering further rational design of new alloys. So far, it has been proposed that the Re effect is rooted in three possible sources: (i) solid solution strengthening of the γ phase due to the large atomic size of Re[1,23], (ii) slowest d-shell diffusion element in Ni[24,25] slowing down dislocation motion, and (iii) the formation of Re clusters within γ acting as strong obstacles to dislocation movement[26,27]. Of these three, mechanism (iii) is the most controversial. Blavette et al.[28,29] found evidence for small Re clusters in γ using a one-dimensional atom probe. On the other hand, more recent observations based on atom probe tomography (APT) show no Re clustering, discounting as such the possible role of Re clustering in improving the creep properties of Ni-based superalloys[23,30,31]. Clearly, further studies are necessary to clarify the possible contribution of Re clustering to the Re effect.

The primary objective of the present work is to advance the understanding of Re effect by exploring Re interaction with structural defects, namely dislocations and SFs in γ′, during creep deformation of single crystal Ni-based superalloys. To this end, observations from advanced scanning transmission electron microscopy (STEM) and near-atomic resolution APT are combined with phase field (PF) modelling of Re enrichment to dislocations during creep. Experimental observations and modelling imply that Re is dragged by dislocations during creep, contributing to the observed effect of Re on creep behaviour. These results suggest the possibility of alloy design through control of solute partitioning to defects.

## Results

**Creep testing and deformation microstructure**. For investigating the effect of Re on the creep response, the single crystal superalloy ERBO/1 is used[32]. As shown in Table 1 in the Methods section, ERBO/1 contains 3 wt% (0.9 at%) Re, and is very close in composition to the commercial alloy CMSX-4. Samples of ERBO/1 were crept at 750 °C and 800 MPa until rupture (~13.5%). Figure 1 shows the detailed creep response and defect evolution of ERBO/1 loaded in [001] direction. An overview of the creep performance is shown in Fig. 1a, b for specimens crept until rupture. The total creep time to rupture is around 530 h. Figure 1b reveals the double-minimum creep behaviour reported previously[33], where an intermediate maximum ($\dot{\varepsilon}_{im}$) at 1% and a global minimum ($\dot{\varepsilon}_{global}$) creep rate at 5% creep strain can be observed. Additional tests were then performed and interrupted at 1 and 5% creep strains.

Conventionally, a creep curve has three stages, i.e., a primary creep stage where the creep rate decreases to a local minimum, a secondary stage where a constant creep rate is reached, and a tertiary stage where the creep rate increases until rupture[1,34,35]. In the present study, before the lowest creep rate was reached, i.e., the global minimum, the creep rate initially increased to an intermediate maximum. As shown in Fig. 1c, it took about 1 h to reach this intermediate maximum creep rate (1% creep strain), and about 172 h to reach the global minimum (5% creep strain). Representative post-creep STEM images of the two interrupted states are displayed in Fig. 1e, f for 1% strain and 5% strain, respectively. Apart from the γ/γ′ microstructures in Fig. 1e, two variants of SFs are imaged. At 1% creep strain, dislocations are mostly confined to the γ channels, with an average dislocation density as high as $3 \times 10^{14}$ m$^{-2}$ in the γ phase[33]. At this stage, the microstructure is heterogeneous, with regions exhibiting a

**Table 1 Elemental composition of ERBO/1 alloy under investigation. Row 2: wt%, Row 3: at%.**

| Co | Cr | Al | Ta | Ti | Mo | Re | W | Hf | Ni |
|---|---|---|---|---|---|---|---|---|---|
| 9.3 | 6.2 | 5.7 | 6.9 | 1.0 | 0.6 | 2.9 | 6.3 | 0.1 | 61 |
| 9.6 | 7.3 | 12.8 | 2.3 | 1.3 | 0.4 | 0.9 | 2.1 | 0.03 | 63.3 |

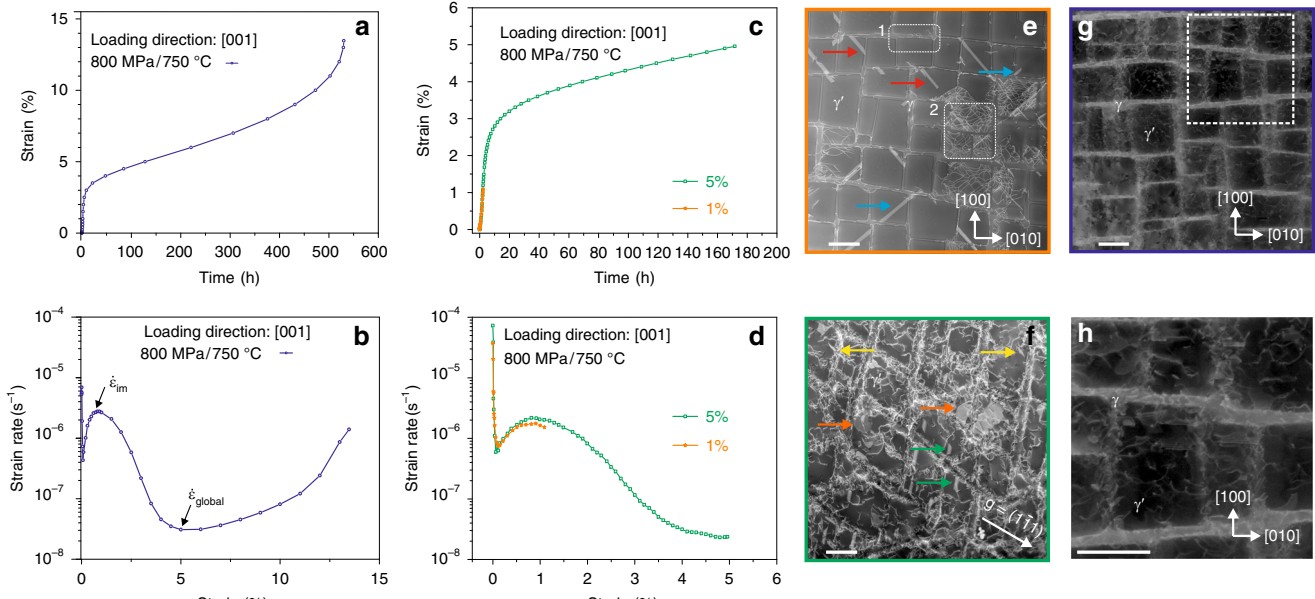

**Fig. 1 Creep curves and microstructures for ERBO-1. Creep condition: 750 °C, 800 MPa, [001] orientation. a** Strain vs. time till rupture. Rupture time: ~530 h. **b** Strain rate vs. strain till rupture. **c** Interrupted creep curves at 1 and 5%, strain vs. time. 1% creep time: ~1 h. 5% creep time: ~170 h. **d** Interrupted creep curves at 1 and 5%, strain rate vs. strain. **e** Representative STEM ADF overview of 1% creep strain, viewed in [001] zone axis. Region 1: region with low dislocation density; region 2: region with high dislocation density. Blue and red arrows show two types of SFs. **f** Representative STEM ADF overview of 5% creep strain under (**g**) condition of (1̄11), showing three different types of SFs, pointed by yellow, orange and green arrows. **g** ECCI overview of ruptured sample showing dislocations in the γ channel and SFs in the γ′ phase. **h** Enlarged view from the rectangle in (**g**). Scale bar is 500 nm.

dislocation density lower than the average, characteristic of early stage deformation (highlighted by the white rectangle labelled '1'). Other regions (shown in rectangle '2') exhibit a much higher dislocation density. Across the microstructure, some of the γ′ precipitates are sheared and SFs can be identified.

In contrast, for the 5% creep strain, the microstructure is altogether more homogeneous, i.e., all three directions of matrix channels are filled with dislocations, and almost every γ′ precipitate is sheared by dislocations and SFs. Previous TEM results have shown that both SISF and SESFs were observed in these two states[33]. When it comes to the ruptured sample, shown in Fig. 1g, h, there is an increasing dislocation density in both, the γ channels and the γ′ precipitates, and neither topological phase inversion nor rafting are observed.

More locally, in Fig. 2, STEM imaging coupled with energy dispersive X-ray spectroscopy (EDS) reveals SISFs and SESFs at both 1 and 5% creep strains within the γ′ precipitates. High resolution STEM (HRSTEM) high angle angular dark field (HAADF) images in Fig. 2a, d, i, j resolve the local atomic structure of SISFs and SESFs, while corresponding STEM-EDS elemental maps of Cr, Co and Re provide insight into local enrichment effects[36,37]. For 1% creep sample, there is no obvious enrichment to either SISFs or SESFs. By comparing the EDX spectrum between the fault region and the precipitate region, as shown in the Supplementary Fig. 1, there is a slight enrichment of Cr and Co for both SISF and SESF. For 5% crept sample, the STEM low angle angular dark field (LAADF) image in Fig. 2h exhibits a region containing a SISF (Fig. 2i) and a SESF (Fig. 2j). A partial dislocation (PD) is observed to be associated with the SISF. The bright contrast in the LAADF image shows the strain contrast, and the corresponding chemical information is obtained in the EDX mapping from Fig. 2k–m.

The EDX mapping results in Fig. 2k–m indicate a clear enrichment of Cr, Co and Re in the SISF. The SESF is also enriched with Cr and Co, as shown in the Supplementary Fig. 2b by comparing the integrated intensity between the defect region and the surrounding precipitate region from the EDS mapping results. The higher intensity of Re in the SESF reveals Re enrichment to SESF. More locally, by comparing the integrated intensity in the PD and SISF shown in the Supplementary Fig. 2, it provides solid evidence of higher concentrations of Cr, Co and Re in the PD compared to the associated SISF. This observation is consistent with the previous work of Smith et al.[19] in another Ni-based superalloy by STEM EDS, where higher Co and Cr concentration is observed at the PD compared to the SESF in γ′.

**Chemical composition analysis.** APT measurements were performed to quantify element enrichment to the SFs. A correlative TEM/APT measurement for the 1% crept sample is shown in Fig. 3. Regions with SFs for correlative TEM/APT specimen preparation are located by using electron channelling contrast imaging (ECCI)[38], as shown in Supplementary Fig. 3. The TEM BF image in Fig. 3a clearly shows the targeted SF in the APT specimen, and the highlighted region represents the area where APT data was acquired. Electron diffraction imaging was used to determine that the fault is extrinsic in nature[37,39], as shown in Supplementary Fig. 4. Figure 3b displays the corresponding APT reconstruction. Co atoms are displayed in dark yellow. Co-enriched γ (shown in darker yellow) and Co-depleted γ′ (shown in lighter yellow) can be clearly distinguished. Inside the γ′ precipitate (pointed by the arrow in Fig. 3b), highlighted by 3.4 at% iso-composition of Cr, a planar feature decorated by Cr locates the SF region.

Figure 3c visualises the same fault edge-on. A Supplementary Movie is provided for a better 3D visualisation. The one-dimensional composition profiles obtained perpendicular to the fault plane (blue arrow) and parallel to the fault through a grooved region (green arrow) across γ and γ′ interface (highlighted by the grey rectangle) are shown in Fig. 3d, e, respectively. Figure 3d shows that the fault region is enriched by approx. 1 at% Co and Cr compared to the surrounding γ′ region, and depleted in Ni and Al. This amount of enrichment is difficult to be directly

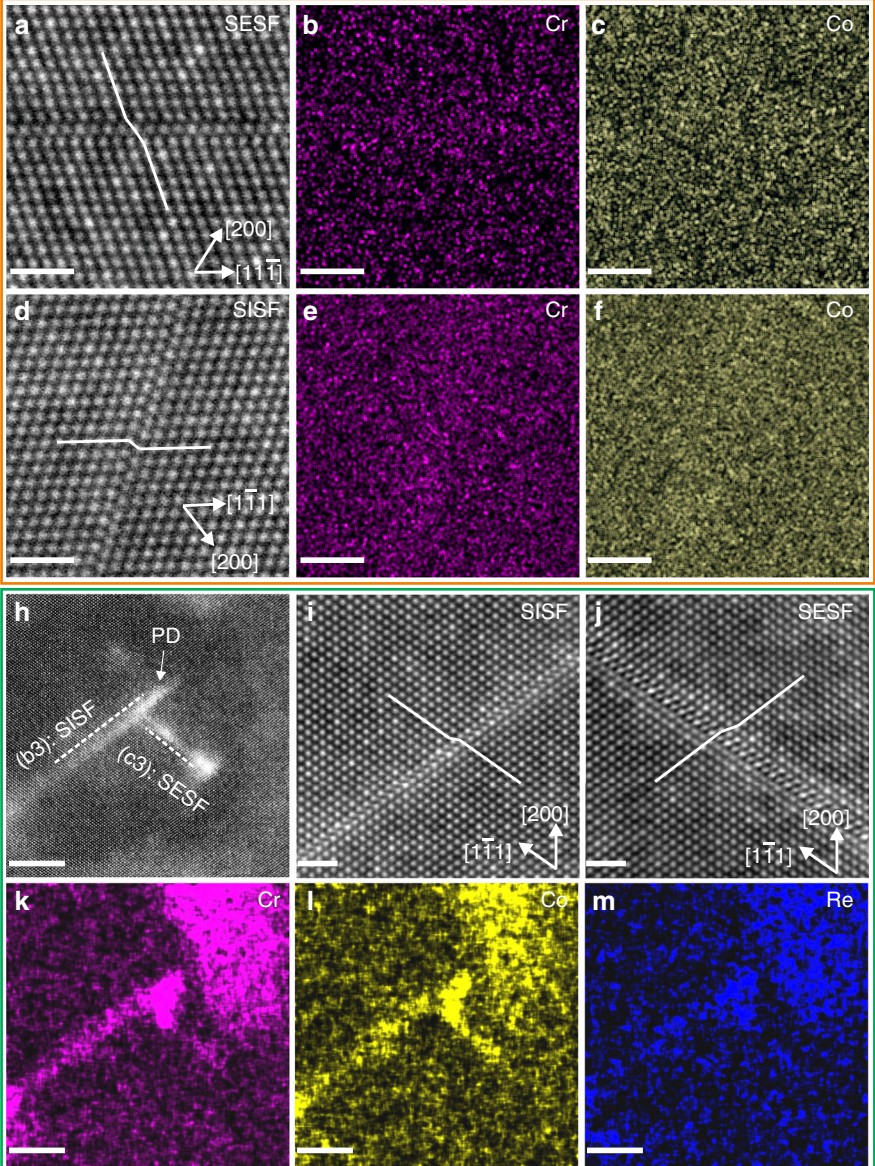

**Fig. 2 Atomic-resolution STEM-EDX of SISF and SESF at 1 and 5% creep strains. 1% crept specimen: a** HRSTEM HAADF image showing the atomic structure of a SESF; **b** Cr EDS mapping for (**a**); **c** Co EDS mapping for (**a**); **d** HRSTEM HAADF image of a SISF; **e** Cr EDS mapping for (**d**); **f** Co EDS mapping for (**d**). 5% crept specimen: **h** STEM LAADF image showing a PD, a SISF and a SESF; **i** HRSTEM HAADF image showing the atomic structure of the SISF; **j** HRSTEM HAADF image showing the atomic structure of the SESF; **k** Cr EDS mapping for (**h**); **l** Co EDS mapping for (**h**); **m** Re EDS mapping for (**h**). Scale bar: 1 nm for (**a–j**); 5 nm for (**k–m**). The colour intensity from EDS mapping is related to the composition in at%. Here, EDS mapping is used to provide a qualitative assessment, and the quantitative assessment comes from the APT measurement shown below.

detected by STEM-EDS, however, the comparison of the spectrum in the fault region (red rectangle) to that in the surrounding γ′ precipitate (blue rectangle) in Supplementary Fig. 1 shows consistent results, i.e., slight enrichment of Co and Cr in the fault region. As for Re, there is no obvious difference between the fault region and the surrounding γ′. However, there is some trace showing a higher Re content in the grooved region (highlighted in Fig. 3f, g), consistent with previous work by Ding et al.[40].

Figure 4 shows results for the APT analysis conducted on the 5% crept sample. Figure 4a displays a 3D reconstruction with Co distribution and a set of iso-composition surfaces delineating regions containing over 4 at% Cr. Two γ′ precipitates can be identified, separated by a large region of γ matrix. In the γ′ precipitate located at the lower part of the reconstruction, a PD is followed by a SF, as shown in a close-up in Fig. 4b. The 2D

projection of the Al at% compositional map in Fig. 4b confirms the planar character of the fault region characterised by an Al deficiency in the fault region to the surrounding γ′ precipitate. Figure 4c, d shows the cut section of the fault plane comparing the distribution of Cr and Re atoms in the γ′ phase, PD, SF and the γ matrix (top: XY plane and side: YZ plane views). Both top and side views show clearly that the leading PD is enriched with higher content of Cr and Re compared to the associated SF, and even more so than the surrounding γ′ precipitate. 1D composition profiles are shown in Fig. 4e–g in the direction perpendicular to the PD and SF plane, reveal a significant enrichment of up to 13 at% in Co and up to 7 at% in Cr in the PD. Even more striking is the increase of Re from ~0.3 at% up to ~1.1 at% in the PD from the γ′ region and up to 0.6 at% in the associated SF. This observation is in line with the STEM EDS mapping analysis in the Supplementary Fig. 2. Regarding the low Re content in the γ′

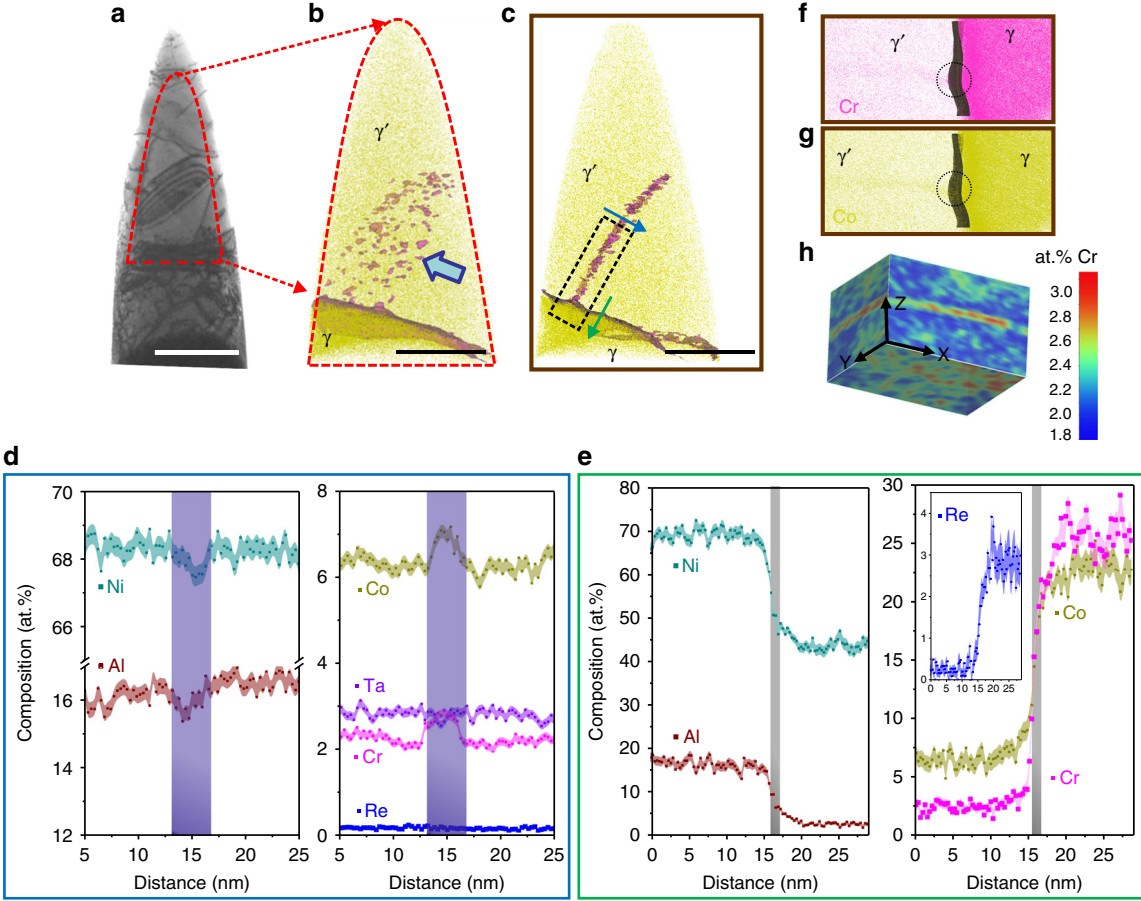

**Fig. 3 Correlative TEM/APT investigation showing a SF from the 1% crept sample. a** TEM BF showing the contrast of a SF. **b** Corresponding APT reconstruction showing a γ/γ′ interface and a SF (highlighted by magenta, 3.4 at% Cr iso-composition surfaces) in the γ′. **c** SF viewed edge-on, indicating the SF is connecting with a trailing partial dislocation in the groove. Two arrows (blue: perpendicular to the SF plane and green: across γ/γ′ interface through the partial dislocation in the groove) show two directions of compositional analysis, the rectangle highlights the fault region. 1D compositional analysis as shown in the two directions: **d** along blue arrow: perpendicular to the fault plane, the light purple rectangle highlights the fault region, and **e** along green arrow: across γ/γ′ interface through the partial dislocation in the groove, and the light grey rectangle shows the γ/γ′ interface. **f** Cr atoms displaying γ & γ′ phases and the SF. **g** Co surface displaying γ & γ′ and the SF and **h** 2D projection in XZ and YZ plane highlighting the fault region. Size of region of interest: $24 \times 12 \times 100$ nm$^3$. Scale bar: **a** 500 nm; **b**, **c** 50 nm. The error bars in Fig. 3d, e are estimated as described in Methods section.

precipitates, the increase of Re in the PD must originate from solutes transported from the γ matrix into γ′, likely via pipe diffusion[11,41–44].

Figure 5 shows APT analysis for another specimen of the same 5% crept sample revealing Re enrichment at γ/γ′ interface dislocations. Figure 5a shows the APT reconstruction with γ/γ′ interfaces separating Co-rich γ and Co-poor γ′ phases. We observe two grooves at the γ/γ′ interfaces marked as G1 and G2. Figure 5b shows the enlarged view of the region containing G1 (dark blue dashed rectangle) with the distribution of Co atoms across the γ/γ′ interfaces. A 2D Re at% compositional map across G1 is also shown, as more clear in Fig. 5c, that indicates a higher Re concentration specifically at the groove relative to the surrounding γ and γ′ phases. The 1D Re at% compositional profile, Fig. 5d, across the groove (along AB in Fig. 5c) reveals an Re enrichment up to ~7.1 at%. This compositional contrast at G1 indicates presence of γ/γ′ interface dislocations with the Re enrichment and will be discussed in detail later.

The compositional analysis by APT for the ruptured sample is shown in Fig. 6. The 3D reconstruction in Fig. 6a shows two γ′ precipitates separated by γ matrix, and the iso-composition surfaces with threshold of 4 at% Cr as γ/γ′ interfaces (in magenta). In the lower right part of the reconstruction in Fig. 6a, there is a planar region showing a SF associated with a PD inside

the γ′ phase. Figure 6b is a close-up of the fault region, and the dashed rectangle highlights the area selected for further compositional analysis. Figure 6d shows the SF and PD in YZ and XZ planes. Composition profiles perpendicular to the SF plane and PD are shown in Fig. 6e, f, respectively. Co (~11 at%) and Cr (~6% at%) contents in the PD are slightly higher than in the SF (Co: ~9 at%, Cr: ~4 at%). These levels of enrichment, compared to γ′, are consistent with those measured in the 5% crept sample. However, there is no Re enrichment observed in both the SF and PD at the ruptured stage. As for the other elements (Ni, Al, Ta, Ti, W, Mo and Hf), their composition profiles are plotted in the Supplementary Fig. 8.

**PF modelling of defect-solute interaction.** In order to gain further insight, PF simulations of the interaction of solute like Re with dislocation and SFs in γ and γ′ have been carried out. Details of the atomistic PF model employed are explained in the Methods section. The PF energy model has been calibrated using the interatomic potential for Ni–Al–Re reported in ref. [45]. For example, in this way, the dependence of the energy of different SFs on the Re concentration in γ and γ′ is determined as presented in Fig. 7a, b. As shown, the ISF energy in γ is reduced when Re is added, favouring Re enrichment to the fault region in

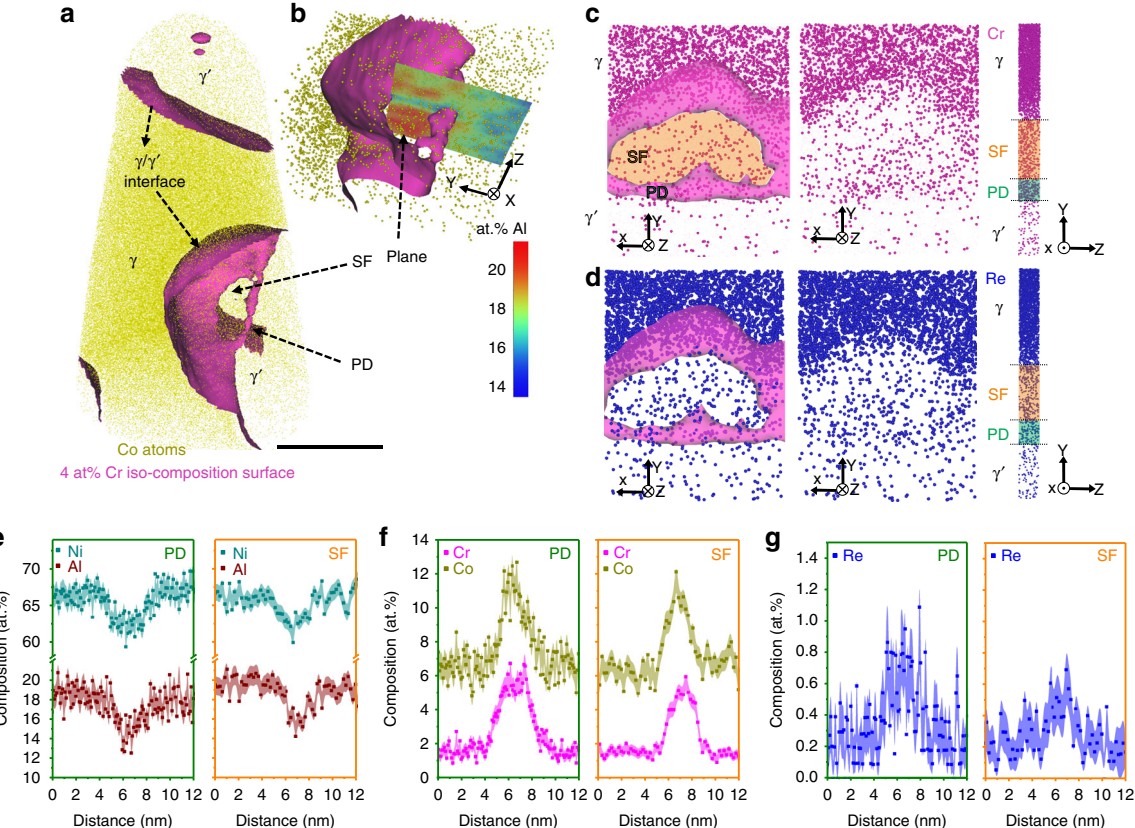

**Fig. 4 APT analysis for 5% creep specimen. a** APT reconstruction showing two γ/γ′ interfaces and a SF in the γ′ precipitate. The SF is connected with a PD. **b** A close-up of the fault region, highlighting with a 2D projection, showing depletion of Al in the fault region. **c** Top (viewed along Z-axis) and side view (viewed along X-axis) showing the distribution of Cr atoms from γ/γ′ interface to SF and PD. **d** Top (viewed along Z-axis) and side view (viewed along X-axis) showing the distribution of Re atoms from γ/γ′ interface to SF and PD. 1D compositional profile along Z showing the distribution of the elements perpendicular to PD (green rectangle) and perpendicular to SF plane (orange rectangle): **e** Ni and Al; **f** Cr and Co; **g** Re. Scale bar in (**a**): 20 nm. The error bars in (**e–g**) are estimated as described in Methods section.

γ[46]. This is also consistent with the fact that Re partitions to the γ phase, as shown in Supplementary Fig. 6, in agreement with the literature (see e.g.,[23,32,47]). On the other hand, in γ′, the fault (i.e., SISF, CSF and APB) energies all increase with the Re concentration. Consequently, these results imply that Re enrichment to faults in the γ′ precipitates is not energetically favoured.

To investigate the Re enrichment to faults and dislocation cores, consider first the case of a static dislocation under no external load inside γ and γ′. The initial Re composition is set to 3.293 and 0.074 at%, in γ and γ′, respectively, based on ThermoCalc[48] equilibrium calculations for ERBO 1 at 750 °C. A $\frac{a}{2}[1\bar{1}0](111)$ and a $\frac{a}{3}[1\bar{1}2](111)$ edge dislocation, is placed in γ and γ′, respectively, and the system is relaxed. Note that both of these dislocation configurations have been previously observed in Ni-based superalloys[49]. During relaxation, the dislocations dissociate as shown in the following manner

$$\gamma : \frac{1}{2}[1\bar{1}0] \rightarrow \frac{1}{6}[2\bar{1}\bar{1}] + \text{ISF} + \frac{1}{6}[1\bar{2}1]. \tag{1}$$

$$\gamma' : \frac{1}{3}[1\bar{1}\bar{2}] \rightarrow \frac{1}{6}[1\bar{1}\bar{2}] + \text{CSF} + \frac{1}{6}[21\bar{1}] + \text{APB} + \frac{1}{6}[\bar{1}2\bar{1}] + \text{SISF}. \tag{2}$$

Simultaneously, Re segregates to the dislocation core and fault regions. As shown in Fig. 7c, d, these simulations predict a maximum Re concentration of 4.6 and 0.088 at% around the dislocation in γ and γ′, respectively. Note that the simulation results are at static equilibrium condition with periodic system

and conserved total amount of solute. Thus, the net amount of Re is constant with dislocation and faults result in redistribution of Re atoms as shown in Fig. 7c, d. Under such conditions for a dislocation without any dragged-in Re to γ′, as reported in Fig. 7d, the model predicts depletion of Re at the fault and maximum segregation of 0.088 to the regions above the dislocation in γ′. As expected, the maximum concentration occurs on the tensile part of the dislocations stress field due to larger size of Re atoms. Since the experimentally observed segregation levels of 0.7–1.0 at% in γ′ are about an order of magnitude larger than the simulation result, this further supports the hypothesis that Re around dislocation cores in γ′ is transported from γ by dislocation drag or pipe diffusion. The former involves transport of Re from γ to γ′ by the drag effect of γ′ shearing dislocations during their movement across γ′ phase. Pipe diffusion[41,50,51] allows solutes to be transported from γ to γ′ phase via dislocations. In addition, the negative Suzuki effect mentioned above results in the depletion of Re in the fault regions in γ′, as shown in Fig. 7d. This is also expected based on the concentration dependency of the fault energies in γ′ as shown in Fig. 7b. The discrepancy between the negative Suzuki effect from simulation and the observed Re enrichment in the SF by APT measurement is discussed in detail in the following section. In summary, given the low Re concentration in the γ′ precipitates and the relatively high concentration of Re in the leading PD, we propose that the experimentally observed enrichment of Re in the PD and SF region in γ′ is due to drag of Re by dislocations and/or pipe diffusion, which is faster than Re diffusion through the

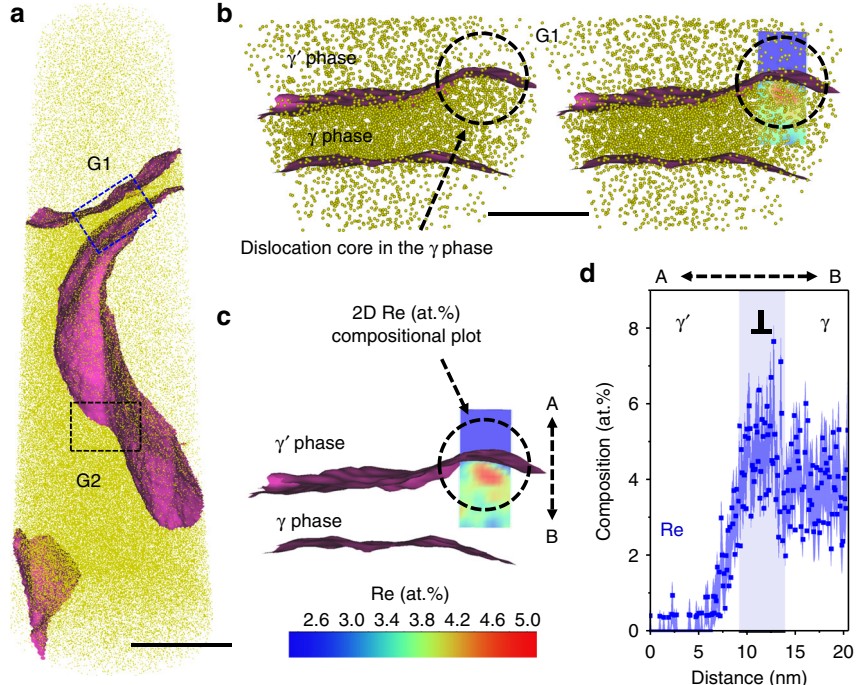

**Fig. 5 APT analysis of γ/γ′ interfacial dislocations for 5% crept sample. a** APT reconstruction showing γ/γ′ interfaces and the two groove regions G1 and G2. **b** Enlarged view of groove region G1 (blue dashed rectangle) with distribution of Co atoms. **c** A 2D Re at% compositional profile across G1 reveals higher Re concentration at G1 indicating Re enrichment at interfacial dislocations. **d** 1D compositional profile of Re across G1 (along AB). Scale bar: **a** 50 nm. **b** 20 nm. The detailed analysis of groove G2 is shown in Supplementary Fig. 5. The error bars in (**d**) are estimated as described in Methods section.

lattice. This is also indicated by the wake of Re left behind by the PD as it moves through γ′.

Note that most dislocations and Re are initially in the γ phase. Thus, knowing how much Re segregates to dislocations in γ is the prerequisite to understand how much Re could be transported by them into γ′. To examine this question in more detail, we consider next combined dislocation glide and Re diffusion in γ. In particular, we consider an edge dislocation subject to an applied shear strain of 1.2%. A Re mobility value of $5.44 \times 10^{-27} \, \text{m}^2 \, \text{mol/J s}$ in γ is assumed based on a diffusion coefficient of $1.67 \times 10^{-21} \, \text{m}^2 \, \text{s}^{-1}$ at $T = 1023 \, \text{K}$ reported in ref. [52]. The thermodynamic relation between diffusion coefficient ($D$) and solute mobility ($M$), i.e., $D = \frac{\partial^2 f}{\partial c \partial c} M$, where $\frac{\partial^2 f}{\partial c \partial c}$, is the second derivative of the free energy density with respect to concentration $c$, is used in this work. The free energy density ($f$) is calculated from the thermodynamic database[52]. Three dislocation mobilities of $2.5 \times 10^{-11}$, $2.5 \times 10^{-10}$ and $2.5 \times 10^{-9} \, \text{m}^3 \, \text{J}^{-1} \, \text{s}^{-1}$ are considered, resulting in average dislocation velocities of about 0.2, 2.0 and 20 nm s$^{-1}$, respectively. Each simulation is carried out without and with Re starting from an initially uniform concentration of 3.293 at%. For simplicity, the dislocation mobility is assumed independent of the solute concentration in the current dilute Re concentration context. The steady-state dislocation/SF configuration in each simulation, as well as the corresponding Re concentration profile, are shown in Fig. 8. The two SF energy maxima shown in Fig. 8a–c are the Shockley partials, and the local minimum between them is the ISF. Beginning with Fig. 8a, note that the energy profile with Re (solid black curve) lags behind (i.e., has moved less than) the energy profile without Re (blue dashed curve) under the same imposed external shear strain and for the same mobility of $2.5 \times 10^{-11} \, \text{m}^3 \, \text{J}^{-1} \, \text{s}^{-1}$. This implies that the Re increase due to segregation shown in Fig. 8a results in a decrease of the dislocation glide (net) driving force, and so the dislocation glide velocity. Note that the solute residual strain is quite large for Re in γ (about 10.6%). This leads to strong interaction between

the dislocation and the Re Cottrell atmosphere. As the dislocation mobility (and so velocity) increases (Fig. 8b, c), Re has less time to be dragged in and segregate, resulting in a lower steady-state Re concentration, and less difference between the energy profiles. In this case, Re diffusion is too slow in comparison to the dislocation velocity, resulting in an elongated solute cloud in the wake of the dislocation.

## Discussion

Figure 9 summarises the main compositional changes during the creep process, especially from 5% creep strain and the ruptured stage. Under loading in the creep process, as shown in Fig. 9a, the interaction of γ matrix/interfacial dislocations leads to the formation of PDs and generation of associated SFs inside of the γ′ phase. The local enrichment of γ matrix elements (Cr, Co and Re) at the interfacial dislocations enables the PDs to carry some of the enriched elements into the γ′ phase. With the evolution of creep strain and longer creep time for segregation, at 5% creep strain, there is an increasing amount of Re at PD and the associated SF inside of the γ′ phase, as shown in Fig. 9b, which is corresponding to a low creep rate state. Upon rupture, only Cr and Co are observed in the PD and SF. In both 5% and ruptured crept specimen, the amount of enrichment is higher in the PD compared to the SF. The detailed evolution of each creep stage is described in the following.

At 1% strain (the early stage of creep at an intermediate maximum creep rate), the microstructure is characterised by SFs in the γ′ precipitates and non-homogeneous distribution of dislocations mostly confined to γ channels, with an average density of $3 \times 10^{14} \, \text{m}^{-2}$ in γ channels and most of the channels are dislocation free. The microstructure reveals deformation occurs by fast moving dislocation ribbons which are switching between γ and γ′ and also changing their configuration. These dislocation ribbons are shown to have a total Burgers vector of $\frac{a}{2}$<112> in γ the phase[49,53]. This stage is also characterised by low

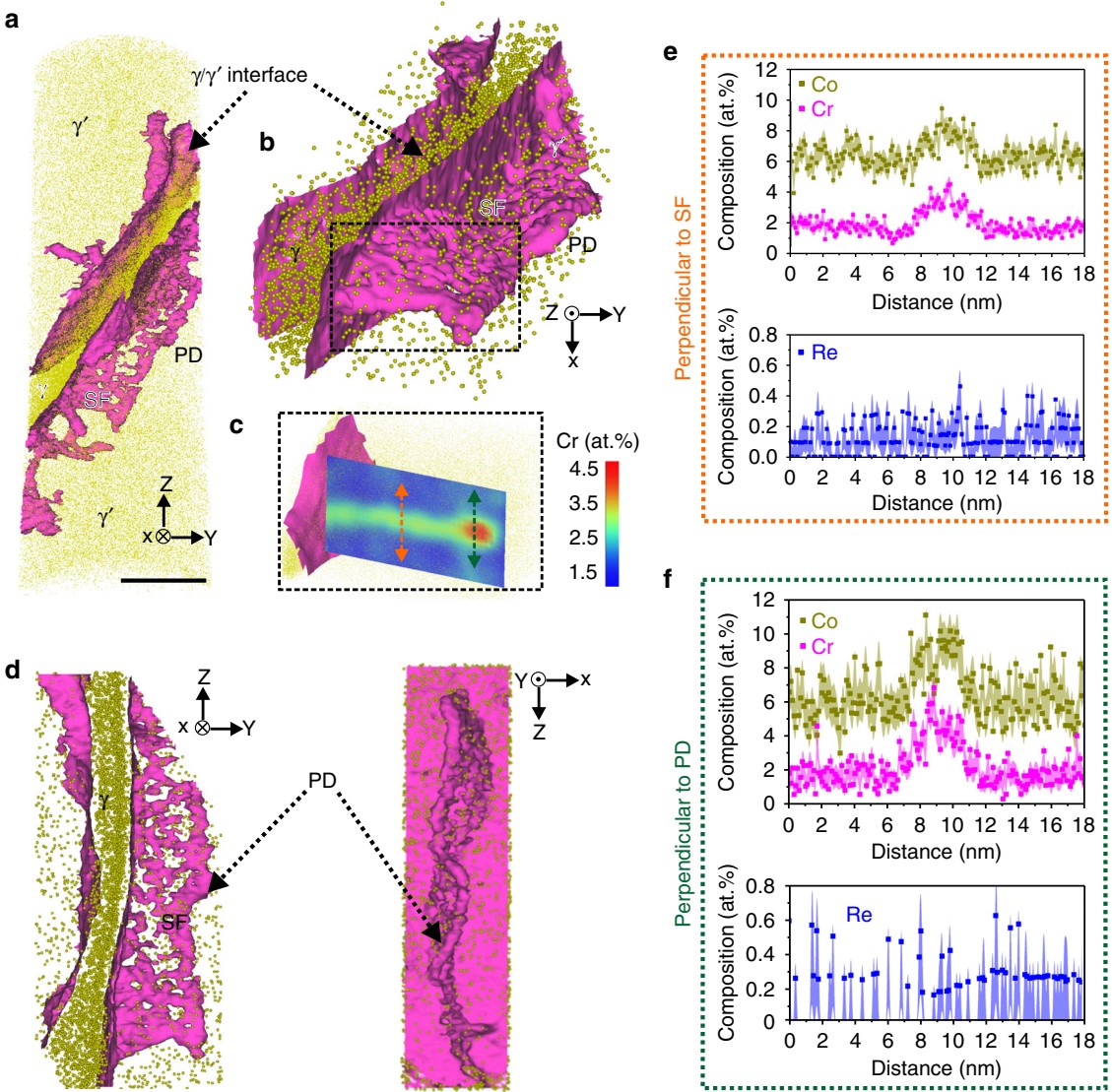

**Fig. 6 APT analysis for the ruptured sample. a** APT reconstruction showing two γ/γ′ interfaces and a SF in the γ′ precipitate. The SF is connected with a PD. **b** Close-up of the fault region and PD. **c** Enlarged view from the dashed rectangle in (**b**), and the 2D projection of Cr highlights that PD is more enriched with Cr compared to the associated SF. **d** Views along X- and Y-axis showing the distribution of 4 at% Cr iso-compositional surface from γ/γ′ interface to SF and PD. **e** 1D compositional profile of Co, Cr and Re perpendicular to SF. **f** 1D compositional profile of Co, Cr and Re perpendicular to PD. Scale bar: **a** 500 nm. The error bars in (**e**, **f**) are estimated as described in Methods section.

concentrations of Cr and Co as well as negligible Re segregation to PDs[51] and SFs in the γ′ precipitates. This initial stage of increasing creep strain rate up to 1% strain (first maximum) can be related to two factors: (1) the continuous generation and movement of dislocations in the γ matrix, with some of the dislocations able to shear the γ′ precipitates, leading to the formation of numerous SFs inside γ′ precipitates; (2) the segregation of Co and Cr to SFs inside γ′, lowering the SFE[54–56] and facilitating further shearing of γ′ by SFs. Our earlier work shows that observed PDs inside the γ′ at 1% creep strain are enriched with Co/Cr and also minor (~0.4 at%) amounts of Re[51]. However, the low amount of Re segregating to dislocations results in negligible drag. Hence, the mechanism of creep at this stage can be assumed to be dominated by the above two factors. This results in the observed increase in creep rate during the initial stage of strain up to 1%.

At 5% strain, the creep rate is two orders of magnitude lower compared to 1%. The microstructure exhibits a high density of dislocations inside the γ channels, which can also impede the

movement of the dislocation ribbons. The presence of SFs in most of the γ′ precipitates show that the deformation is dominated by the shearing of individual γ′ precipitates, and it has a higher activation energy[53,57] than the primary creep stage at 1% strain. In addition, the compositional analysis (Figs. 2 and 4) of structural defects shows Re enrichment at PDs and SFs inside the γ′ precipitates. The decrease in creep rate in this second stage is due to both structural and compositional effects. First, the increased dislocation density and dislocation interactions in γ lead to work hardening and a reduced creep rate. Compared to the primary creep (1%), the activation energy in secondary creep stage (5%) is significantly higher, as measured and discussed in ref.[53,57]. The high activation energy raises the importance of dislocation climb at γ/γ′ interface during the creep process. Intensive dislocation activity in γ channels leads to the distorted γ/γ′ interface, which changes the local stress state of the interface and the interfacial dislocations. This change of the stress state, further facilitate Re accumulation at the γ/γ′ interface (Fig. 5). In turn, the accumulated Re at the γ/γ′ interface can thus act as an effective barrier for

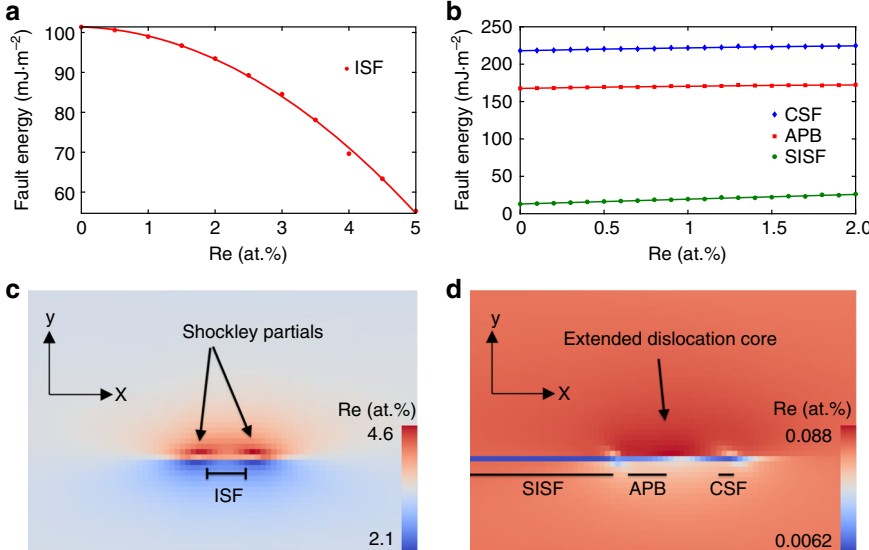

**Fig. 7 Phase-field calculation of fault energies and segregation profiles.** Fault energies in γ (**a**) and γ′ (**b**) as a function of Re concentration calculated using the interatomic potential in ref. [45]. Point data and solid lines represent the atomistic results and the fitted model used in PF, respectively. Re concentration field around a dissociated edge dislocation in γ (**c**) and γ′ (**d**). The dislocation glide plane normal is along *y* and Burgers vector direction is along *x*. The initial uniform Re concentration was set to 3.293 and 0.074 at% in γ and γ′, respectively.

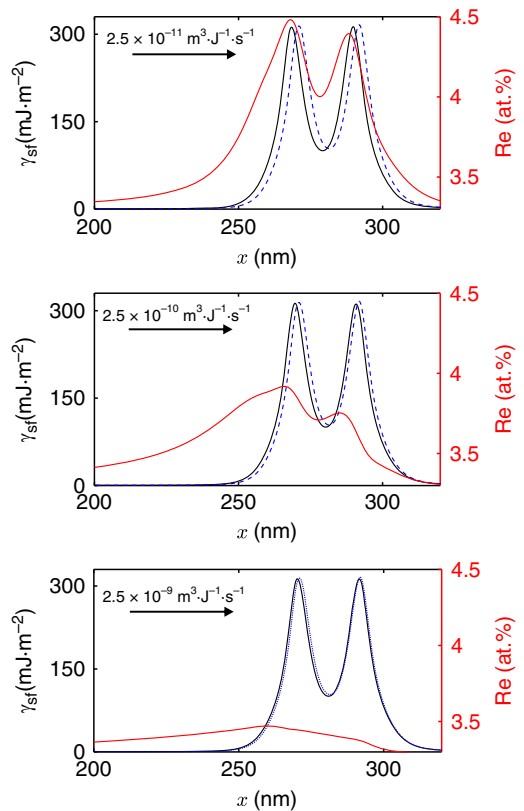

**Fig. 8 Phase-field of dislocation/SF energy and composition profiles for moving dislocations.** Snapshots of dislocation/SF energy profiles with (solid black line) and without (dashed blue line) Re as well as the corresponding Re concentration (solid red line) in γ. The direction of dislocation motion is marked by arrows together with the average dislocation mobilities (velocities): **a** $2.5 \times 10^{-11}$ m³J⁻¹s⁻¹ (0.2 nm s⁻¹); **b** $2.5 \times 10^{-10}$ m³ J⁻¹ s⁻¹ (2 nm s⁻¹) **c** $2.5 \times 10^{-9}$ m³J⁻¹s⁻¹ (20 nm s⁻¹).

dislocation climb[23] and hence helps resist γ′ shearing. Simultaneously, a longer creep duration (~172 h) provides more time for Re to enrich the interfacial dislocations as well as more time for more dislocations to glide from γ into γ′. This will lead to enhanced concentration of Re to PDs and SFs in γ′. From 1 to 5% creep strain, progressive enrichment of Re to these PDs results in a net reduction in dislocation (glide driving force and so) velocity, enabling further Re enrichment. This would explain the higher Re concentration in PDs at 5% strain.

The Re enrichment in PDs and the resulting decrease in their velocity is believed to be an essential contributing factor to creep rate reduction and so an increased creep resistance during this stage. To estimate such an influence of Re on dislocation motion, a simplified calculation of the dislocation velocity and the associated Re-Cottrell atmosphere mobility is carried out: the dislocation velocity can be estimated using the Orowan equation $\dot{\varepsilon} = b \cdot \rho_m \cdot v_d$, where $\dot{\varepsilon}$ is the strain rate, $b$ is Burger's vector magnitude, $\rho_m$ is the mobile dislocation density, and $v_d$ is the average dislocation density. Since the strain rate at 1% creep strain is around $2.35 \times 10^{-6}$ s⁻¹, taking a mobile dislocation density of dislocation density from γ′ ($0.9 \times 10^{13}$ m⁻²)[33] yields in this way a dislocation velocity of about 1–2 nm s⁻¹. The same calculation at 5% creep strain with a strain rate of $2.23 \times 10^{-8}$ s⁻¹ and a γ′ dislocation density of $3.0 \times 10^{13}$ m⁻² yields a velocity two and a half orders of magnitude lower, with a velocity of ~$5 \times 10^{-3}$ nm s⁻¹. This difference in velocity between 1% creep strain and 5% creep strain also implies that at 1% creep strain, the higher creep rate and corresponding higher average dislocation velocity results in less Re enrichment and weaker solute drag to the dislocations (consistent with PF simulation results of fast-moving dislocation in Fig. 8c). In contrast, at 5% creep strain, the lower creep rate and corresponding lower average dislocation velocity facilitates increased solute segregation and a stronger Re drag force on the dislocation motion, comparable to the slow-moving dislocation in PF simulation results in Fig. 8a.

The velocity of a Re Cottrell atmosphere can be estimated using the approach proposed by Titus et al.[18] and other researchers[13,16]. For 1.2 at% Re and a diffusion rate of $1.67 \times 10^{-21}$ m² s⁻¹, one obtains a Re atmosphere velocity of $4.25 \times 10^{-2}$ nm s⁻¹. The detailed calculation is provided in the Supplementary Note 1.

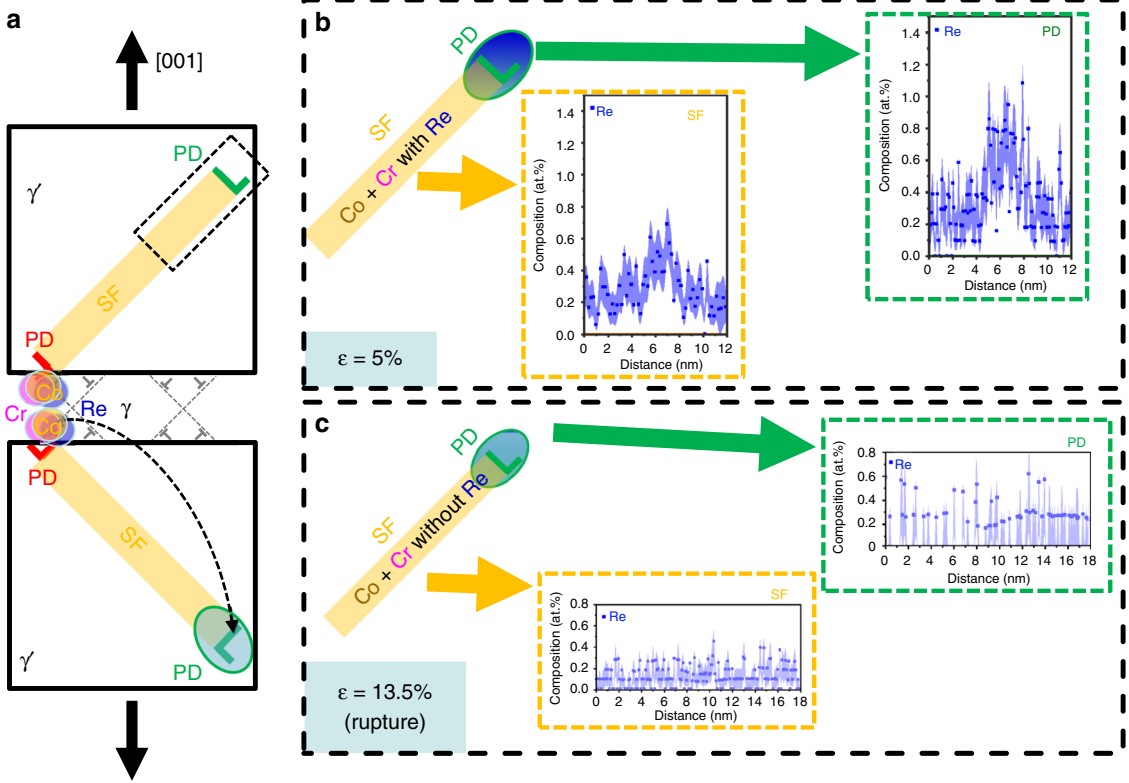

**Fig. 9 Schematic evolution of element enrichment (i.e., Re) at SF and PDs during creep. a** Schematic showing the generation of SFs by PDs inside of γ′ phases with segregation in the dislocations in the γ phase. Further analysis focuses on the dashed rectangle region of a PD and associated SF. **b** 5% creep strain: Re Cottrell atmosphere around PD and less enrichment of Re in the SF. **c** Rupture: decreased amount of Cr and Co segregating to the PDs compared to the 5% creep stage, no Re enrichment observed and less dragging effect at this stage. More information see text.

While the velocity of the Cottrell atmosphere of other solutes, such as Cr and Co[13,16,18], is higher than 50 nm s$^{-1}$, Re as the slowest diffusing element in Ni is the rate-limiting factor for the creep process. The discrepancy of the dislocation velocity and of the Re Cottrell atmosphere indicates a strong dragging effect (dragging dislocations from further movement) by Re acting on PDs.

At 5% creep strain, the observation of Re enrichment to SFs indicates active transport of solutes by PDs, very likely by pipe diffusion, or along SFs, as they cut into the γ′ precipitates, carrying elements (Re, Cr and Co) from the γ matrix. The presence of solutes at PDs can first result from local partitioning[58], the extent of which is mostly unknown for most solutes at such defects. Current PF modelling of Re enrichment to SFs results in an increase in the SFE in γ′ (Fig. 7b), and so little effect of Re concentration on the driving force for Re enrichment to SFs, conversely to the leading PD, i.e., it is energetically not favourable for Re to segregate to SFs. Yet Re is observed at SFs, likely because its diffusivity does not allow it to keep up with the dislocation motion. With Re segregating to PDs in γ′ at 5% creep strain, if we assume that every PD carries approx. 1.2 at% Re, and based on the measured dislocation density, up to one-third of the Re atoms are transported from the γ to the γ′ phase (see Supplementary Note 2). After 5% creep strain, the depletion of the Re content in γ will also increase the SFE of γ, which allows easier dislocation cross slip and weakens its work hardening capabilities[59,60]. In the meantime, the depletion of Re in the γ matrix after 5% creep strain leads to a reduced content of Re as sources to segregate to dislocations during further creep deformation.

At the point of rupture, the creep rate is proportional to the creep strain, and both the mobile dislocation density and the dislocation velocity increase. The progressive increase in strain

rate can be ascribed either to the generation of new mobile dislocations or to an acceleration of the movement of existing dislocations. The possible appearance of cavitation and pores relaxes the stress, but upon further loading may result in local stress concentration which can free dislocations pinned by Cottrell atmospheres or increase their velocity such that less Re enrichment is possible. Faster dislocations can result in an increase in creep rate. Thus, till the stage of rupture, the increasing difference between the dislocation velocity and Re diffusivity, as well as relatively lower Re content leads to a weaker effect of Re dragging dislocations, as supported by Fig. 8, which explains the increase of creep rate at this stage.

In conclusion, we studied the creep behaviour of a Ni-based single crystal superalloy ERBO/1 at a low temperature (750 °C) and high stress (800 MPa) state, and found a correlation between the level of enrichment of Re to PDs and SFs in the γ′ precipitates and the creep rate. Re is present up to over 1 at% in the PD at 5% creep strain, along with Co and Cr. At the early stage (1% creep strain) and at the ruptured stage, when the strain rate is much higher, less Co and Cr segregation to the PDs in the γ′ precipitates, and no Re enrichment, is observed. Static and dynamic PF simulations show that Re enrichment to dislocations results in a reduction of the driving force for dislocation glide. Assuming that the corresponding increase in Re concentration does not increase the dislocation mobility, this will result in a reduced dislocation glide velocity. In this case, Re segregating to dislocations, as observed here in γ′, would lead to a reduced creep rate in addition to any such reduction coming for example from work hardening induced by dislocation interactions. Our present observations further the understanding of creep in Ni-based superalloys, in particular the effect of Re on dislocation motion, and provide direct evidence of solute drag and/or pipe diffusion

in dislocations, resulting in transport of matrix elements into the γ′ precipitates. Such additional insight into the Re effect may offer guidelines for designing next generation superalloys based on which elements segregate to mobile dislocations. For example, by adjusting the alloy composition in other heavy elements (e.g., W, Ru, Ta, etc.), it might be possible to slow down the creep process in the early stages, thereby further increasing creep life.

## Methods

**Uniaxial tensile creep**. Miniature dog-bone tensile specimen was cut by electron discharging machining from the bulk plates with a composition shown in Table 1. Miniature specimen with a gauge length of 9 mm was used for uniaxial tensile creep testing at 750 °C under a high stress of 800 MPa in the orientation of [001]. Creep test till rupture, along with interrupted tensile testing at a strain of 1 and 5% were conducted for a reproductive measurement. A creep machine from Denison Mayers was utilised, which was equipped with three zone furnace for a precise control of creep temperature. During the constant load creep test, the specimens were heated to 750 °C under a preload of 20 MPa, and the required stress was reached within few seconds. When desired strain was reached, the specimen was quickly cooled down to around 400 °C by compressed air and air-cooled till room temperature. The whole cooling process took around 30 min. The specimens remained under load during the whole cooling process, for minimising the effect of cooling process upon the deformation microstructure and chemistry.

**Electron microscopy**. Scanning electron microscopy (SEM) and transmission electron microscopy (TEM) were employed for a detailed microstructure evolution investigation. ECCI in the scanning electron microscope[61] was used for identification of specific feature within its large field-of-view, and guide site-specific preparation of correlative TEM/APT measurements. ECCI measurement was carried out using a Zeiss Merlin SEM from Carl Zeiss SMT AG, equipped with a Gemini-type field emission gun electron column and a Brucker e-flash HR EBSD detector. TEM overview and HRSTEM was conducted using a FEI Titan Themis 60–300 S/TEM. It was equipped with a high-brightness field emission gun, gun monochromator and Super-X EDX detector operated at 300 kV. STEM imaging was carried out with a Fischione Instruments (Model 3000) high angle annular dark field detector (HAADF). The STEM-HAADF image is obtained at a camera length of 100 mm, and the STEM–LAADF image is taken with a camera length of 245 mm. For combined elemental information, STEM elemental maps employing energy dispersive spectroscopy (EDS) were recorded with a semi-convergence angle of 23.8 mrad and a camera length of 100 mm, which corresponds to an inner and outer semi-collection angle of 73 and 200 mrad, respectively. STEM EDS mapping is acquired using both Brucker Esprit and Velox software for 5–10 min, to avoid contamination and get sufficient count.

**Atom probe tomography**. Sample preparation for APT was carried out mainly by a dual-beam focused ion beam (FIB) instrument (FEI Helios Nanolab 600i), for both routine APT measurement (tips mounted on Si coupon)[62] and a correlative TEM/APT measurement (tips mounted on half-Mo-grid)[38,63]. The lift-out specimen was sharped to a tip radius smaller than 100 nm, followed by a low kV shower for removing Ga damage. A LEAP 5000 XR from Cameca Instruments Inc.: was used for the measurement of precise chemical composition, operated in laser pulsing mode. For a better resolution of the elements detected, a pulse repetition rate of 125 kHz and a pulse energy of 50 pJ was used, while the specimen was kept at a base temperature of 60 K. The detection rate was kept at a frequency of 1 ion per 100 pulses on average. At least three specimens with targeted defects were prepared and analysed for each creep state. The error bars in the 1D compositional analysis was calculated as $2\sigma = \sqrt{\frac{C_i(1-C_i)}{N}}$, where $C_i$ represents the composition of each solute $i$ and $N$ represents the total number of atoms in each bin.

**PF modelling**. Atomistic PF modelling of dislocations[64,65] and its recent extension to include solute diffusion[66,67] is used to model dislocation interaction with Re in Ni-based superalloys. A simplified ternary model alloy consisting of Ni–Al–Re is considered. The γ phase is modelled as a disordered (Ni, Re) FCC lattice and γ′ phase is modelled as an ordered Ni₃(Al, Re) phase[68]. The energetic part of PF model for dislocation and solute diffusion is calibrated using the Ni–Al–Re embedded atom interatomic potential from[45] as well as CALPHAD[48]. For both γ and γ′, elastic stiffness, solute residual distortion, as well as the dislocation and SFEs, are all determined as a function of Re concentration and used to identify or calibrate the model free energy. Based on the low expected equilibrium and segregated Re concentration in each phase, the concentration dependence of these model parameters is determined up to 5 at% and 2 at% in γ and γ′ phases, respectively. Note that the PF energy model accounts in particular for energy storage in both stacking faults and dislocation cores. As in the atomistic case, stacking faults and dislocation cores (lines) are resolved independently by the PF model. Since the PF energy also depends on chemical composition, segregation affects both core and fault morphology (e.g., their sizes). More details about the PF

model are reported in ref. [67]. LAMMPS[69] is used for atomistic calculations based on the Ni–Al–Re potential[45]. All PF simulations are carried out in DAMASK[70].

## Data availability

The data that support the findings of this study are available from the corresponding authors upon reasonable request.

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

## Acknowledgements

X.W., S.K.M., B.G., D.R., D.B. and G.E. are grateful for funding from the Deutsche Forschungsgemeinschaft (DFG) through Project A1, A2 and A4 of the collaborative research center SFB/Transregio 103 superalloy single crystals. SKM also acknowledges financial support from the Alexander von Humboldt Foundation. B.G. acknowledges financial support from the ERC-CoG-SHINE-771602. J.R.M., P.S. and B.S. acknowledge financial support of the Deutsche Forschungsgemeinschaft (DFG) in Sub-project M5 of the Priority Programme 1713 "Chemomechanics". PS thanks the EPSRC for financial support through the associated programme grant LightFORM (EP/R001715/1). The authors are grateful to U. Tezins and A. Sturm for their technical support of the atom probe tomography and focused ion beam facilities at the Max-Planck-Institut für Eisenforschung. Dr. Y. Chen and Dr. P. Kontis are kindly acknowledged for their fruitful discussions.

## Author contributions

X.W. generated the idea and designed the experiments. X.W. and S.K.M. carried out the APT experiments. S.K.M., X.W. and BG analysed the APT data. X.W. and C.L. performed the STEM analysis. J.R.M., P.S. and B.S. conducted phase field modelling, D.B. carried out the creep tests. G.E., D.R. and G.D. supervised the project. X.W. and B.G. wrote the paper and all authors contributed to the discussion and revision of the paper.

## Competing interests

The authors declare no competing interests.
