## [Peer Review File · Nature Communications]

Reviewers' Comments:

Reviewer #1:

Remarks to the Author:

This is an important paper contains some novel and very significant results, namely the observation of the segregation of Re to faults and dislocation during the low temperature creep of a Ni-base superalloy. I have however some misgivings about the interpretation of these results and the story this tells about the mechanisms involved.

Specifically;

1. The degree of segregation to the PD and the fault for the 5% creep sample, shown in figures 5 and again in figure 9, claim a segregation level of 1-2% at the PD (lines 238 and 318) and 0.8% in the SF (line 237). This is shown in Figure 9 as a line drawn through the results which shows a peak in a very noisy data set greatly in excess of the maximum single value measured. I would estimate that a line drawn through the dataset in Figure 5b peaks at around 0.7% (PD) and at about 0.5 for Figure 5c (SF). Their case for Re segregation is strong but is not improved by exaggeration.
2. On page 13 the results of the phase-field modelling are presented and Figure 7d shows negative segregation to the fault plane itself, (as you would expect as it raises the SFE in the model potential used) and an enhancement above the dislocation and depletion below. The text, line 317, quotes the level of segregation to the fault as 0.088, this being the level on the upper plane, but is not taking into account the depletion below. Thus, the PF model is predicting exactly what you would expect, that the effect of the stress field of the edge dislocation would shift larger atoms such as Re from the compressive side of the dislocation to the tensile. It would be useful to evaluate whether the model predicts a net accumulation of Re at the fault which is what is being measured experimentally. Interestingly two of the three dislocation cores show depletion and the third is invisible. It is not clear from the method section how the core of the dislocation is dealt with in the PF model.
3. The comment on page 14, line 318 that the increased Re concentration supports the hypothesis that Re is transported through the dislocation without explaining where it is going or why. Hence the need to demonstrate that there is a net increase in the Re content around the dislocation in the gammaprime. Unless this is the case, the dislocation could be slowed simply by the need for the Re to reposition due to the passage of the dislocation, provided the velocity was slow enough to accommodate this.
4. Page 16, line 363: This sentence makes no sense; it says in effect: due to the change in local Re concentration there is no Re at the fault. The authors need to clarify what they mean.
5. Page 17: The discussion from line 376 onwards does not recognize that there are significant differences between the deformation at 1% strain and 5% strain in addition to the strain rate and dislocation density. At 1% most of the dislocation channels remain dislocation free and the deformation occurs by fast moving dislocation ribbons which are switching between gamma and gammaprime and also changing their configuration. Often the dislocation ribbon retains a Burgers vector of $a/2 \langle 112 \rangle$ in the gamma phase in this early stage. This part of the deformation also has a distinctly lower activation energy than later stages. During secondary creep which is occurring at the 5% used here most of the gamma channels contain significant dislocation density impeding movement of the ribbons and although the deformation remains due principally to the dislocations in the gammaprime it is on a precipitate by precipitate basis with a higher activation energy more typical of creep at higher temperatures. It is argued that this is due to the Re in the gamma restricting climb in the gamma/gammaprime interface necessary to form SF able to penetrate the gammaprime. This would be consistent with the high Re observed at the gamma/gammaprime interface in Figure S5. [ref 1]

6. Page 17 Line 396 and Figure S5page 32; The authors do not discuss the widely accepted argument that during cooling from the test temperature the gamma prime will grow and that this will reject Re from the gamma consumed resulting in a bow-wave of Re unable to diffuse away during the rapid cooling. Specimens that fail sometimes experience different cooling rates if they fall from the hot zone – confirmation that this is not the case here would be significant. [ref 2]

7. The increased dislocation density at the gamma/gamma prime interface in the mature deformation structure is relieving the stress due to misfit rather than increasing it. Again, segregation to the tensile side of the dislocation would lower energy and has been seen to distort the shape of the gamma/gamma prime interface at dissociated dislocations. [ref 3]

8. The levels of Re in the gamma channels is assumed to drop as the Re accumulates at the interfaces

Overall the paper presents some fascinating results but the discussion has some significant problems which require clarification. The level of Re segregation is significant but not as high as they claim in the text. The authors suggest that the dislocations in the gamma prime are acting rapid diffusion paths for Co, Cr and Re, but do not explain why Re is diffusing to faults where it raises the energy. They need to confirm that the PF model requires a net diffusion of Re.

Minor changes to text:

Line 296: not energetically favoured.

Line 363: which corresponds to a low creep rate state

Line 439: does not allow it to keep up with ...

Line 475: which elements

Refs:

1. A. Mottura et al. Acta Materialia 58 (2010) 931–942

2. G.L. Drew, R.C. Reed, K. Kakechi, C.M.F. Rae, , Proceedings of the 10th International Symposium for Superalloys, ed. K.A. Green, T.M. Pollock, H. Harada, T.E. Howson, R.C. Reed, J.J. Schirra, and S Walston, The Minerals, Metals and Materials Society (TMS) Warrendale, PA, USA, 127-136 (2004)

3. V.A. Vorontsov, L. Kovarik, M.J. Mills, C.M.F. Rae, (2012), Acta. Mat., 60, 12, 4866-4878.

Reviewer #2:

Remarks to the Author:

This is an interesting article dealing with the role of Re in the creep performances of nickel base superalloys. These researches are of utmost importance for applications for turbine blades in aeronautics.

The main interest comes from the combination of high-resolution analytical microscopes (TEM, APT) with mechanical tests and phase field modelling. The manuscript is rather well written but there is some times repetitions. The discussion is interesting but suffers from a lack of clarity on some aspects dealing with the concept of equilibrium segregation to lattice defects, solute drag, pipe diffusion. This article is thought to deserve to be published provided some corrections and clarifications are brought up on the following items:

1. There is a persisting ambiguity in the manuscript when authors talk about segregation. APT and TEM show solute enrichment (Co,Cr,Re) to lattice defects such as SF or dislocations, but this does not give the proof that it is solute segregation. The observed enrichment of Re to dislocations could be the result of the swifter diffusion of Re along dislocation lines, as discussed by the author in their manuscript. This can be nothing but a snapshot of the diffusion short circuits. If this is the

case, authors should not call this segregation of Re but simply Re enrichment to line defects that is the fact that is observed. Writing segregation is ALREADY an interpretation of facts.

2. The enrichment rates that are observed are very small both for Co, Cr but also for Re (a small factor 3 between Re in the dislocation core and the surrounding solid solution, fig. 5). This would lead to segregation energies (as given by Mc Lean equation or Cottrell theory) very very small. It would be interesting to get an order of magnitude of this using thermodynamical predictions (e.g. Mc lean equation). In agreement with phase field studies, this would suggest that solute enrichment does not really lower energies of lattice defects in γ' precipitates. This could also be the case for dislocations in γ . Predictions (line 314) also predict very faint segregation effects (0.088 at.% of Re in the dislocation!?) leading to the conclusion that enrichments to dislocation are nothing but snapshots of pipe diffusion effects.

3. What about drag effects? More details on the related physics should be given (interaction of migrating solutes with the stress field of moving dislocations...). Are they likely to occur in this context where interactions of Re with dislocations are very small? This should be discussed. The opinion of authors on that is not very clear. Drag effects versus pipe diffusion effects should be discussed.

4. How are you sure that the SF is really connected with a dislocation (fig. 6). Is there a proof that enrichments are really located on a dislocation as APT images does not show any evidence of the presence of such a defect (atomic planes are not shown). Is it only TEM images in correlation that strongly suggest that?

5. Line 332 : are you using the Darken equation $D=MRT$? If right , you could recall it.

6. Line 292: Phase field indicates that Re decreases the SF energy in γ but experiments show very faint effects (fig. 5(c)). This should be discussed.

7. Fig 9: precise that figure are related to SDF and dislocation in (b) and (c)

8. Line 385 : precise what is really new in this article compared to article [49]

9. Line 409: As this equation is simple, remind the Orowan equation.

10. Line 440 : add creep strain after 5%

11. Line 443 : put space after γ

12. Line 453 : You are talking about Cottrell atmospheres but it appears that the observed Re enrichment at dislocation lines are not really segregation effects but pipe diffusion effects (cf. previous questions). Again, this should be clarified (question 1). See also line 461.

13. Line 528 : error bars are incorrect as it should be 2σ . In addition, when local concentrations are considered, the standard deviation is smaller and σ has not the same expression but should include the detector efficiency (see article of F. Danoix et al. (Ultramicroscopy, Volume 107, Issue 9, September 2007, Pages 739-743)

Reviewers' comments

Reviewer #1 (Remarks to the Author):

This is an important paper contains some novel and very significant results, namely the observation of the segregation of Re to faults and dislocation during the low temperature creep of a Ni-base superalloy. I have however some misgivings about the interpretation of these results and the story this tells about the mechanisms involved.

We appreciate the reviewer for finding the present work to be novel and significant. We also thank for the constructive comments and suggestions to improve the manuscript especially on the proposed mechanisms. The point by point response to the comments is as follows.

Specifically;

1. The degree of segregation to the PD and the fault for the 5% creep sample, shown in figures 5 and again in figure 9, claim a segregation level of 1-2% at the PD (lines 238 and 318) and 0.8% in the SF (line 237). This is shown in Figure 9 as a line drawn through the results which shows a peak in a very noisy data set greatly in excess of the maximum single value measured. I would estimate that a line drawn through the dataset in Figure 5b peaks at around 0.7% (PD) and at about 0.5 for Figure 5c (SF). Their case for Re segregation is strong but is not improved by exaggeration.

We thank the reviewer for the comment and agree that the line drawn to represent the trend of the Re profiles in Figures 9b and 9c appear exaggerated, compared to the actual profiles shown in Figures 5b and 5c. We have now modified Figure 5 and Figure 9 to incorporate new profiles from the same acquired APT data set and adjusted the bin size in the profile to lower the noise level. We do hope that the reviewer will find the modified Re plots more convincing.

2. On page 13 the results of the phase-field modelling are presented and Figure 7d shows negative segregation to the fault plane itself, (as you would expect as it raises the SFE in the model potential used) and an enhancement above the dislocation and

depletion below. The text, line 317, quotes the level of segregation to the fault as 0.088, this being the level on the upper plane, but is not taking into account the depletion below. Thus, the PF model is predicting exactly what you would expect, that the effect of the stress field of the edge dislocation would shift larger atoms such as Re from the compressive side of the dislocation to the tensile. It would be useful to evaluate whether the model predicts a net accumulation of Re at the fault which is what is being measured experimentally. Interestingly two of the three dislocation cores show depletion and the third is invisible. It is not clear from the method section how the core of the dislocation is dealt with in the PF model.

We thank the reviewer for the comment. Indeed, the reported Re segregation level of 0.088 at.% is the maximum concentration around the dislocation in γ' . As expected, the maximum concentration occurs on the tensile part of the dislocations stress field due to larger size of Re atoms. Note that these simulation results are under static equilibrium condition with periodic boundary conditions and a conserved total amount of solute. Thus, the net amount of Re in the simulation is constant. As the reviewer mentioned, the dislocations and fault result in a redistribution of the Re atoms as shown in Figure 7c and d. Under static conditions for a dislocation without any dragged-in Re into γ' , as reported in Figure 7d, the model predicts depletion of Re at the fault and maximum segregation of 0.088 to the regions above the dislocation. Since the experimentally observed segregation levels are about an order of magnitude larger that can be attributed as the dislocations are either dragging the Re atoms into γ' , or the dislocations act as hollow pipe, providing a path for fast Re diffusion. Since there are many differences between the model and the experiments, this is of course not the only possibility. Clarification and further discussion along these lines has been added to the revised manuscript on Page 14 in the *Results* section.

Regarding the modeling of the dislocation core, the PF energy model accounts in particular for energy storage in both stacking faults and dislocation cores. As discussed in the paper, the entire energy is calibrated using an EAM potential and CALPHAD data. As in the atomistic case, stacking faults and dislocation cores (lines) are resolved independently by the PF model. Since the PF energy also depends on the chemical

composition, segregation affects both core and fault morphology (e.g., their sizes). Additional details concerning the PF model were added to the revised manuscript on Page 24 in the *Methods* section.

3. The comment on page 14, line 318 that the increased Re concentration supports the hypothesis that Re is transported through the dislocation without explaining where it is going or why. Hence the need to demonstrate that there is a net increase in the Re content around the dislocation in the gamma prime. Unless this is the case, the dislocation could be slowed simply by the need for the Re to reposition due to the passage of the dislocation, provided the velocity was slow enough to accommodate this.

We thank the reviewer for pointing out the necessity for clarification. Here, the experimentally observed Re concentration (0.7 - 1 at.%) at the dislocations inside γ' phase is much higher than it is predicted by phase field simulations (0.088 at.%). As mentioned in the response to the previous comment, the simulation results shown in Figure 7d are for Re segregation to a static dislocation in γ' with a conserved total amount of solute. In this case, the γ phase is the only source of Re. We thus propose that the higher Re concentration at the dislocation core might be due to two solute mass transport mechanisms, i.e., either by dislocation drag and/or pipe diffusion. As the amount of Re in dislocations observed in γ' is larger compared to the calculated value, it is possible that the observed Re is dragged in by γ dislocations into γ' . Since the simulation results are for segregation to a static dislocation, they are relevant to the case that Re diffusion/segregation is much faster than dislocation motion. Pipe diffusion is another possibility where dislocations act as hollow pipe [Kolbe et al., MSEA 1988; Legros et al., Science 2008; Wu et al., Materialia 2018] transporting solute along the dislocation core from γ to γ' . This is discussed in more detail in the revised manuscript on Page 13 in the *Results* section.

4. Page 16, line 363: This sentence makes no sense; it says in effect: due to the change in local Re concentration there is no Re at the fault. The authors need to clarify what they mean.

We agree with the reviewer and now the sentence has been removed to avoid further confusion.

5. Page 17: The discussion from line 376 onwards does not recognize that there are significant differences between the deformation at 1% strain and 5% strain in addition to the strain rate and dislocation density. At 1% most of the dislocation channels remain dislocation free and the deformation occurs by fast moving dislocation ribbons which are switching between gamma and gamma prime and also changing their configuration. Often the dislocation ribbon retains a Burgers vector of $a/2 \langle 112 \rangle$ in the gamma phase in this early stage. This part of the deformation also has a distinctly lower activation energy than later stages. During secondary creep which is occurring at the 5% used here most of the gamma channels contain significant dislocation density impeding movement of the ribbons and although the deformation remains due principally to the dislocations in the gamma prime it is on a precipitate by precipitate basis with a higher activation energy more typical of creep at higher temperatures. It is argued that this is due to the Re in the gamma restricting climb in the gamma/gamma prime interface necessary to form SF able to penetrate the gamma prime. This would be consistent with the high Re observed at the gamma/gamma prime interface in Figure S5. [ref 1]

We are grateful to the reviewer for this important part of discussion that we had left out of our initial explanation. Now we have modified the entire paragraph including reviewer's valuable points and suggestions. The modified section is as follows:

“At 1% strain (the early stage of creep at an intermediate maximum creep rate), the microstructure is characterized by SFs in the γ' precipitates and non-homogeneous distribution of dislocations mostly confined to γ channels, with an average density of $3 \times 10^{14} \text{ m}^{-2}$ in γ channels and most of the channels are dislocation free. The microstructure reveals deformation occurs by fast moving dislocation ribbons which are switching between γ and γ' and also changing their configuration. These dislocation

ribbons are shown to have a total Burgers vector of $a/2\langle 112 \rangle$ in γ the phase [52,53]. This stage is also characterized by low concentrations of Cr and Co as well as negligible Re segregation to PDs [50] and SFs in the γ' precipitates. This initial stage of increasing creep strain rate up to 1% strain (first maximum) can be related to two factors: 1) the continuous generation and movement of dislocations in the γ matrix, with some of the dislocations able to shear the γ' precipitates, leading to the formation of numerous SFs inside γ' precipitates; 2) the segregation of Co and Cr to SFs inside γ' , lowering the SFE [54–56] and facilitating further shearing of γ' by SFs. Our earlier work shows that observed PDs inside the γ' at 1% creep strain are enriched with Co/Cr and also minor (~ 0.4 at.%) amounts of Re [50].”

“At 5% strain, the creep rate is two orders of magnitude lower compared to 1%. The microstructure exhibits a high density of dislocations inside the γ channels, which can also impede the movement of the dislocation ribbons. The presence of SFs in most of the γ' precipitates show that the deformation is dominated by the shearing of individual γ' precipitates, and it has a higher activation energy [52,57] than the primary creep stage at 1% strain. Additionally, the compositional analysis (Figures 2 and 4) of structural defects shows Re enrichment at PDs and SFs inside the γ' precipitates. The decrease in creep rate in this second stage is due to both structural and compositional effects. First, the increased dislocation density and dislocation interactions in γ lead to work hardening and a reduced creep rate. Compared to the primary creep (1%), the activation energy in secondary creep stage (5%) is significantly higher, as measured and discussed in [52,57]. The higher activation energy raises the importance of dislocation climb at γ/γ' interface during the creep process. Intensive dislocation activity in γ channels leads to the distorted γ/γ' interface, which changes the local stress state of the interface and the interfacial dislocations. This change of the stress state further facilitates Re accumulation at the γ/γ' interface (Figure 5). In turn, the accumulated Re at the γ/γ' interface can thus act as an effective barrier for dislocation climb [23] and hence helps resist γ' shearing. Simultaneously, a longer creep duration (~ 172 h) provides more time for Re to enrich the interfacial dislocations as well as more time for more dislocations to glide from γ into γ' . This will lead to enhanced concentration of Re to PDs and SFs in γ' .

From 1% to 5% creep strain, progressive enrichment of Re to these PDs results in a net reduction in dislocation (glide driving force and so) velocity, enabling further Re enrichment. This would explain the higher Re concentration in PDs at 5% strain.”

6. Page 17 Line 396 and Figure S5 page 32; The authors do not discuss the widely accepted argument that during cooling from the test temperature the gamma prime will grow and that this will reject Re from the gamma consumed resulting in a bow-wave of Re unable to diffuse away during the rapid cooling. Specimens that fail sometimes experience different cooling rates if they fall from the hot zone – confirmation that this is not the case here would be significant. [ref 2]

We thank the reviewer for the comment.

In the present work, to minimize the effect of cooling and maintain post creep microstructure, the crept miniature specimen was firstly cooled down by compressed air to around 400 °C and followed by air cooling till room temperature. This has been described in the methods section. We would also like to mention that none of our specimens, used for microstructural characterisation, were failed and fell from the hot zone.

Additionally, we also observe grooves at γ/γ' interfaces without any evidence of solute segregation, especially Re, in the present investigated alloy. For example, in the same dataset shown in Figure 5, we also observe groove G2 at the γ/γ' interface (black checked box) and its compositional analysis is shown below as Figure R1. Similar 1D and 2D Re compositional profiles across the groove G2 reveals no Re enrichment with respect to γ phase (R1c and d). Hence, the Re compositional analysis at G1 (modified Figure 5) represents the enrichment of Re at interfacial dislocations. Figure R1 is also added to the revised manuscript as supplementary figure S6.

Figure R1: APT analysis of grooves at the interface for 5% crept sample. (a) APT reconstruction showing γ/γ' interfaces and two groove regions G1 and G2. The detailed analysis of groove G1 is shown in Figure 5. (b) Enlarged view of G2 groove region with distribution of Co atoms and (c) 2D Re at.% compositional profile across G2 showing no enrichment of Re with respect to γ phase. (d) 1D compositional profile across G2 (AB) of Re. Scale bar: (a) 50 nm. (b) 20 nm. (Supplementary figure S6 in the revised manuscript).

7. The increased dislocation density at the gamma/gamma prime interface in the mature deformation structure is relieving the stress due to misfit rather than increasing it. Again, segregation to the tensile side of the dislocation would lower energy and has been seen to distort the shape of the gamma/gamma prime interface at dissociated dislocations. [ref 3]

We agree with the reviewer comments and the valuable suggestion. Now we have modified the text, page 18, in particular reference to the work [ref. 3]. We also modified the supplementary figure S5 and mentioned, in accordance with the reviewer suggestion, that similar γ/γ' interface grooves/distortions have been observed in 5% crept specimen, as shown in the STEM HAADF Figure R2 below. We see clearly the serrated interface and SSF shearing the γ' precipitate.

Figure R2: STEM HAADF images for 5% crept specimen showing serrated γ/γ' interface. (a) Low magnification showing several serrated positions at the γ/γ' interface with SFs. (b) enlarged view showing one example of the SSF connecting with the serrated position. (Supplementary figure S5 in the revised manuscript). Scale bar in (a): 50 nm; (b) 5 nm.

8. The levels of Re in the gamma channels is assumed to drop as the Re accumulates at the interfaces.

We agree with the reviewer's comment that Re concentration in the γ channels drops as the Re accumulates at the γ/γ' interfaces. We have included a simplified estimation of the amount of Re carried away from γ into γ' in supplementary material S2. In addition, the reduced level of Re in the γ after 5% creep deformation leaves less source of Re for dislocations to pick up at later stage of deformation.

Overall the paper presents some fascinating results but the discussion has some significant problems which require clarification. The level of Re segregation is significant but not as high as they claim in the text. The authors suggest that the dislocations in the gamma prime are acting rapid diffusion paths for Co, Cr and Re, but do not explain why Re is diffusing to faults where it raises the energy. They need to confirm that the PF model requires a net diffusion of Re.

We thank the reviewer for the appreciation of present results and constructive suggestions/comments for improving the discussion. We have now modified the figures

with better representation of Re segregation/enrichment effects and modified the text accordingly. The discussion on the dislocations acting as rapid diffusion paths for the solutes has also been included in the modified manuscript. We have extended the discussion on Re segregating to the superlattice stacking faults. In support to the atomic scale experimental measurements at different creep strain rate levels, we observe that Re segregation behaviour at the PDs and its associated SFs varies significantly. However, as the reviewer pointed out and also shown in Figure 7 by phase field simulation, Re segregation to SISF increases its energy and hence it is not expected Re residing at the fault plane. We thus would like to clarify and propose that Re segregation to the faults is more related to a dynamic/kinetic effect. As also illustrated in Figure 8, Re segregation to the fault can be triggered due to the differences between the PD velocity and Re diffusivity with higher creep strain. More specifically, the slow moving Re is rather left behind by the fast-moving shearing PD, and hence Re traces were found experimentally at the faults created by the PDs. Figure 8 demonstrates clearly the effect of dislocation mobilities on the Re concentration at the dislocations and its associated fault behind. We have now also included a brief description related to this proposed mechanism.

Minor changes to text:

Line 296: not energetically favoured.

Line 363: which corresponds to a low creep rate state

Line 439: does not allow it to keep up with ...

Line 475: which elements

We thank the reviewer for pointing these out. The sentences have been revised as suggested and highlighted in the manuscript.

Refs:

1. A. Mottura et al. *Acta Materialia* 58 (2010) 931–942
2. G.L. Drew, R.C. Reed, K. Kakechi, C.M.F. Rae, *Proceedings of the 10th International Symposium for Superalloys*, ed. K.A. Green, T.M. Pollock, H. Harada, T.E. Howson,

R.C. Reed, J.J. Schirra, and S Walston, The Minerals, Metals and Materials Society (TMS) Warrendale, PA, USA, 127-136 (2004)

3. V.A. Vorontsov, L. Kovarik, M.J. Mills, C.M.F. Rae, (2012), Acta. Mat., 60, 12, 4866-4878.

Reviewer #2 (Remarks to the Author):

This is an interesting article dealing with the role of Re in the creep performances of nickel base superalloys. These researches are of utmost importance for applications for turbine blades in aeronautics.

The main interest comes from the combination of high-resolution analytical microscopes (TEM, APT) with mechanical tests and phase field modelling. The manuscript is rather well written but there is some times repetitions. The discussion is interesting but suffers from a lack of clarity on some aspects dealing with the concept of equilibrium segregation to lattice defects, solute drag, pipe diffusion. This article is thought to deserve to be published provided some corrections and clarifications are brought up on the following items:

We thank the reviewer for the positive response and constructive comments on our work which we address individually in what follows.

1. There is a persisting ambiguity in the manuscript when authors talk about segregation. APT and TEM show solute enrichment (Co,Cr,Re) to lattice defects such as SF or dislocations, but this does not give the proof that it is solute segregation. The observed enrichment of Re to dislocations could be the result of the swifter diffusion of Re along dislocation lines, as discussed by the author in their manuscript. This can be nothing but a snapshot of the diffusion short circuits. If this is the case, authors should not call this segregation of Re but simply Re enrichment to line defects that is the fact that is observed. Writing segregation is ALREADY an interpretation of facts.

We thank the reviewer for the comment. As the reviewer points out, segregation underlines thermodynamic equilibrium condition, which is not discussed/investigated in the present work. We thus correct the term segregation to enrichment accordingly throughout the manuscript.

2. The enrichment rates that are observed are very small both for Co, Cr but also for Re (a small factor 3 between Re in the dislocation core and the surrounding solid solution, fig. 5). This would lead to segregation energies (as given by Mc Lean equation or Cottrell theory) very very small. It would be interesting to get an order of magnitude of this using thermodynamical predictions (e.g. Mc lean equation). In agreement with phase field studies, this would suggest that solute enrichment does not really lower energies of lattice defects in γ' precipitates. This could also be the case for dislocations in γ . Predictions (line 314) also predict very faint segregation effects (0.088 at.% of Re in the dislocation!!?) leading to the conclusion that enrichments to dislocation are nothing but snapshots of pipe diffusion effects.

We thank the reviewer for the comment. Indeed, the model predicts an increase in fault energies in γ' as a function of Re concentration. This results in depletion of Re in the fault regions as shown in Figure 7d. The results shown in this figure are for the case of a dissociated static dislocation in bulk γ' with initially zero Re decoration. Redistribution of Re in such cases (due to the core stress field and concentration-dependent fault energies) shows maximum concentration of 0.088 at.% just above the dislocation in the tensile part of the stress field. Since this value is well below the experimentally observed Re concentration around the dislocation in γ' that can be attributed as in the experiments, the dislocations are either dragging the Re atoms into γ' , or dislocations act as hollow pipe, providing fast path for Re diffusion. Clarification and more discussion along these lines has been added to the revised manuscript on Page 14.

3. What about drag effects? More details on the related physics should be given (interaction of migrating solutes with the stress field of moving dislocations...). Are they likely to occur in this context where interactions of Re with dislocations are very small?

This should be discussed. The opinion of authors on that is not very clear. Drag effects versus pipe diffusion effects should be discussed.

We thank the reviewer for the comment on the dislocation drag effects. Figure 8 and the discussion in page 15 elaborately describes the effect of Re solute on the dislocation glide in γ phase by phase field simulations. We observe a strong Re drag effect on the dislocation motion. The solute residual strain is rather large for Re in γ (about 10.6%). This leads to strong interactions between the dislocation and the Cottrell atmosphere of Re. Now, we have also added a discussion on the interaction of Re and the dislocation core. Additionally, we also discussed on the solute drag and pipe diffusion effects in page 14 of the revised manuscript.

4. How are you sure that the SF is really connected with a dislocation (fig. 6). Is there a proof that enrichments are really located on a dislocation as APT images does not show any evidence of the presence of such a defect (atomic planes are not shown). Is it only TEM images in correlation that strongly suggest that?

We comply with the reviewer that a prior TEM image of the obtained APT data set in figure 4 could have strongly supported the made claims. But, we were not able obtain the TEM image of for particular sample. These samples were creep ruptured with a high density of defects and thus were prone to frequent fracture during field evaporation by APT. However, we would like to invite the reviewer to refer to earlier extensive correlative TEM/APT work from our group in similar multicomponent superalloy (Figures 3 and 4 in Ref [13], Figure 1 and 2 in Ref [12]) that shows a clear chemical and structural contrast between an SSF and its associated partial dislocation by APT and HRSTEM respectively. Hence, with reference to the morphological appearance of the defect features as per the mentioned correlative work, here, we clearly identified similar compositional contrast between a SF and its associated PD (Figure 6c) by APT as illustrated by a 2D Cr iso-compositional surface. More details on the compositional contrast can be found in the description of Figure 6 in the main text.

5. Line 332: are you using the Darken equation $D=MRT$? If right, you could recall it.

We thank the reviewer for the comment. We use the thermodynamic relation between diffusion coefficient (D) and solute mobility (M), i.e. $D = \frac{\partial^2 f}{\partial c \partial c} M$, where $\frac{\partial^2 f}{\partial c \partial c}$ is the second derivative of the free energy density with respect to concentration c , calculated from thermodynamic database in Ref [49]. This is explained in the revised manuscript on Page 15.

6. Line 292: Phase field indicates that Re decreases the SF energy in γ but experiments show very faint effects (fig. 5(c)). This should be discussed.

We thank the reviewer for the comment. Phase field results in Figure 7 shows Re decreasing SFE of γ phase. This proves that Re prefers to segregate to SFs in γ phase. Supplementary figure S6 (previous S5) shows higher concentration of Re in γ phase compared to γ' phase. Figure 4(c and d), however, manifests the local enrichment of Cr, Co and Re in the PD and SF in γ' phase. To clarify this, the description from line 292 is updated as following:

“As shown, the ISF energy in γ is reduced when Re is added, favouring Re segregation to the fault region in γ [46]. This is also consistent with the fact that Re partitions to the γ phase, as shown in supplementary figure S6, in agreement with the literature (see e.g. [23,32,47]).”

7. Fig 9: precise that figure are related to SDF and dislocation in (b) and (c)

We thank the reviewer for the comment. Figure 9(b) and (c) has been updated accordingly.

8. Line 385: precise what is really new in this article compared to article [49]

We thank the reviewer for the comment. In the previous work (Ref. [49]), we show only the evidence of Re enrichment (as well as Cr and Co) at the partial dislocations inside of γ' and proposed that this observation is related to pipe diffusion of solutes along the dislocation core.

Here though, we went much further than this preliminary work, and we have finally rationalised the “Re effect”, i.e. provided the missing link between observations in multiple states and the mechanical lifetime of the material. This required the combination of electron microscopy and atom probe tomography, but also the phase field modelling, which was not previously used.

We have more systematically performed detailed atomic scale defects analysis at different creep strains. The insights into the structure of the defects and their local composition lead to, for the first time, the precise characterisation of the actual distribution of Re atoms across the individual defect features (PD and SF) inside the γ' phase. The variation of the Re composition at these defect features as a function of the creep strain, supported by phase field simulation, unveils the specific role of Re on creep deformation of Ni-based single crystal superalloys. We do hope, the reviewer will find it more convincing regarding the novelty of the present work in superalloy metallurgy.

9. Line 409: As this equation is simple, remind the Orowan equation.

We thank the reviewer for the comment. The Orowan equation has been added and highlighted in the manuscript. The detailed calculation using Orowan equation is shown in the supplementary material S1.

10. Line 440 : add creep strain after 5%

We thank the reviewer for the comment. “creep strain” has been added accordingly.

11. Line 443 : put space after γ

We thank the reviewer for the careful reading. Space has been added after γ .

12. Line 453 : You are talking about Cottrell atmospheres but it appears that the observed Re enrichment at dislocation lines are not really segregation effects but pipe diffusion effects (cf. previous questions). Again, this should be clarified (question 1). See also line 461.

We thank the reviewer for the comment. We have now modified the concerned text on comparison of solute drag effect and pipe diffusion along dislocation cores in discussion section, page 14.

13. Line 528 : error bars are incorrect as it should be 2σ . In addition, when local concentrations are considered, the standard deviation is smaller and σ has not the same expression but should include the detector efficiency (see article of F. Danoix et al. (Ultramicroscopy, Volume 107, Issue 9, September 2007, Pages 739-743)

We thank the reviewer for the correction and we have now incorporated it. We agree with the reviewer's point regarding the standard deviation for inclusion of detector efficiency. However, we calculated the standard deviation with the formula mentioned in the method section, which tends to maximise the error estimation. The error bars we use are hence a more conservative approach, since introducing the considerations on the detector efficiency, as done by Dr Frédéric Danoix would narrow down the associated compositional fluctuations.

Reviewers' Comments:

Reviewer #1:

Remarks to the Author:

The authors have taken on board most of the points made and I am happy with the changes made. I have picked up the following.

Line 23: Re *enrichment*

Line 35, factors *to* further..

Figure 1 (g) wrongly placed white square

Line 319 and 321 *respectively* repeated, ..dislocation are placed in gamma and gamma' respectively, ...

Line 361: conce*n*tration

Reviewer #2:

Remarks to the Author:

The manuscript can now be accepted for publication

REVIEWERS' COMMENTS:

Reviewer #1 (Remarks to the Author):

The authors have taken on board most of the points made and I am happy with the changes made. I have picked up the following. Line 23: Re *enrichment* Line 35, factors *to* further.. Figure 1 (g) wrongly placed white square Line 319 and 321 *respectively* repeated, ..dislocation are placed in gamma and gamma' respectively, ... Line 361: conce*n*tration

Reply: We sincerely thank the careful review from the reviewer. All the mentioned points have been corrected in the manuscript with the tracked changes.

Reviewer #2 (Remarks to the Author):

The manuscript can now be accepted for publication 10

Reply: We sincerely appreciate the reviewer from the reviewer and are happy that the reviewer is satisfied with the replies.